# Photovoltaic Thermal Collectors Integrated with Phase Change Materials: A Comprehensive Analysis

Muthanna Mohammed Awad [1], Omer Khalil Ahmed [1], Obed Majeed Ali [2], Naseer T. Alwan [1,3], Salam J. Yaqoob [4,*], Anand Nayyar [5], Mohamed Abouhawwash [6,7] and Adel Fahad Alrasheedi [8,*]

1   Kirkuk Technical College, Northern Technical University, Mosul 41002, Iraq; muth80ma@gmail.com (M.M.A.); omerkalil@yahoo.com (O.K.A.); nassir.towfeek79@gmail.com (N.T.A.)
2   Renewable Energy Research Unit, Northern Technical University, Kirkuk 36001, Iraq; obedmajeed@gmail.com
3   Department of Nuclear Power Plants and Renewable Energy Sources, Ural Federal University, 620002 Yekaterinburg, Russia
4   Department of Research and Education, Authority of the Popular Crowd, Baghdad 10001, Iraq
5   Faculty of Information Technology, Graduate School, Duy Tan University, Da Nang 550000, Vietnam; anandnayyar@duytan.edu.vn
6   Department of Mathematics, Faculty of Science, Mansoura University, Mansoura 35516, Egypt; abouhaww@msu.edu
7   Department of Computational Mathematics, Science, and Engineering (CMSE), College of Engineering, Michigan State University, East Lansing, MI 48824, USA
8   Department of Statistics and Operations Research, College of Science, King Saud University, Riyadh 11451, Saudi Arabia
*   Correspondence: engsalamjabr@gmail.com (S.J.Y.); aalrasheedi@KSU.EDU.SA (A.F.A.)

**Abstract:** The target of the current study was to review and analyze the research activities of previous studies on cooling techniques for thermal photovoltaic (PV) systems using phase-change materials. These materials have the ability to absorb and release certain amounts of potential heat energy by changing their state from phase to phase (solid–liquid) within a small temperature range. These materials have been used to regulate and lower the temperature, increase the efficiency, and extend the life of solar cells. A host of improvements have been made to phase-changing materials through the combined utilization of phase-change materials and fins in addition to nanoscale fluids to enhance electrical efficiency. When using PCMs, the thermal, electrical, and overall efficiency improved by 26.87%, 17.33%, and 40.59%, respectively. The addition of nanomaterials increased phase-change materials' specific heat capacity and thermal conductivity, thus reducing the plate temperature and increasing the electrical efficiency. It was found that using of nanoparticles together with a microcapsule had better performance in terms of energy efficiency. Studies indicated that variable phase materials were not used because of their high cost and lack of stable operational design. Therefore, the effect of phase-change materials on PV/thermal (PVT) system performance needs further investigation and study.

**Keywords:** photovoltaic; solar energy; phase-change materials; nanofluids; photovoltaic thermal (PVT)

## 1. Introduction

Renewable energy is one of the most appealing options for decreasing global energy requests, especially in the heating and cooling sectors, where nonfossil sources account for less than 20% of total energy consumption [1]. Increasing fossil fuel usage leads to a rise in air pollution and the earth's temperature, causing global warming, a major threat to a human lifetime [2]. Solar energy can be considered a viable source of renewable energy that is characterized by its availability in most countries of the world and can be easily exploited without the need for advanced technology. Solar energy systems are also distinguished by their low operating costs and ease of maintenance. Solar energy systems are divided into solar collectors, to heat water or air, and solar cells, to generate electricity [3]. The

most important disadvantages of solar energy are its lack of continuity due to sunset and its need for energy storage systems, which cause additional costs [4]. However, solar energy harvested by solar photovoltaic systems is the most plentiful, inexhaustible, and environmentally friendly form of energy generation [5]. Different types of photovoltaic (PV) systems, which turn solar power into usable energy, are available. In general, the two major types of PV systems used today can be summarized as PV panels and hybrid photovoltaic thermal (PVT) systems [6]. Solar photovoltaic systems and solar collectors can generate only electrical or thermal energy. However, hybrid PVTs system can produce thermal and electrical energy together simultaneously [7]. PV cells are semiconductors that transform direct current (DC) discharged and concentrated solar radiation with conversion efficiency varying from 4 to 32%, depending on the properties of the material from which they are manufactured [8]. The wide use of solar cells faces many challenges, and the most important obstacle is their low efficiency resulting from their high temperature. In reality, most arriving solar radiation is transformed into heat, raising the PV panel's temperature and lowering the output power, electrical conversion efficiency, and fill factor [9]. With every 1 °C rise in solar panel temperature, the generation efficiency of a standard crystalline-silicon solar panel decreases by 0.45%, as shown in Figure 1 [10]. It is also desirable to increase PV module heat dissipation whenever possible [11].

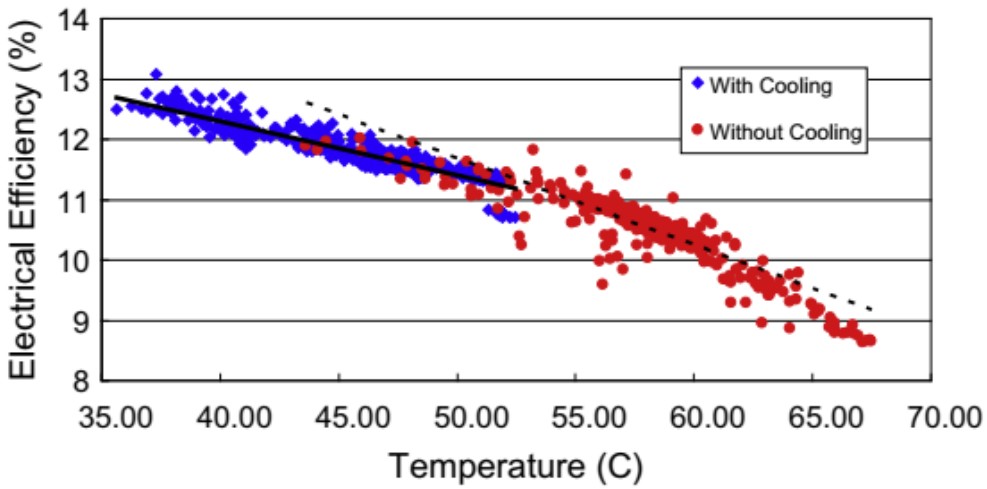

**Figure 1.** The relationship between solar cell efficiency and temperature [10]. Adapted with permission from Elsevier.

In recent years, some researchers have discussed the applications of PCM systems in PV modules, as they have been reported for cooling PV cells or modules [12] efficiently. PCMs have a few simple advantages in comparison with cooling using forced or natural convection of air: higher heat-absorbing performance with no mobile equipment and low maintenance costs and electrical consumption. A few attempts were made to incorporate PCMs into PV panels, producing unique PVT panels [13]. There are many ways to integrate solar voltaic systems with PCM that are reviewed herein, and the simplest of these methods is shown in Figure 2 [14]. Containers containing phase-change material are placed on the solar panel's back surface and fixed there. The harmful excess heat from the solar cells is transferred to be stored in the phase-changing matter in the form of latent heat. The application of materials to absorb the excess heat released by PV modules is the basis for passive cooling strategies. As opposed to active cooling methods, PCM integration on the rear of a PV module is a preferred passive cooling solution because of the lower operation and maintenance costs [15]. In addition, in dry and hot climates, the photoelectric temperature rises above the recommended temperature limit for operation, which ranges between 40 and 85 °C, resulting in decreased efficiency and reduced working life for PV cells. This necessitates a control for the PV temperature [16]. Passive cooling by exploiting

PCMs is an innovative alternative to efficiently cool PV modules without using forced cooling and consuming additional energy [17].

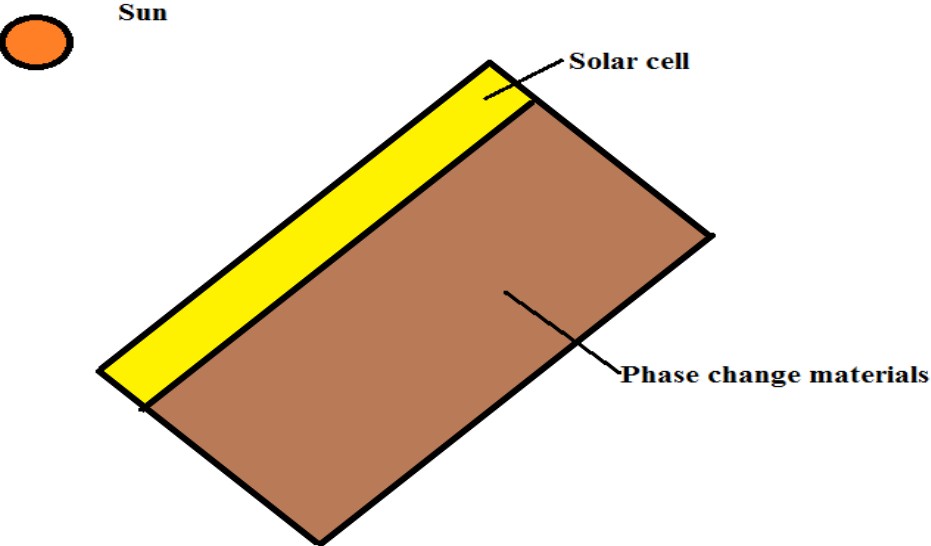

**Figure 2.** The basic structure of PCMs in PV application.

Phase-change material is very interesting for thermal environmental control and storage of energy [18]. It efficiently stores and releases a huge quantity of latent heat during the processes of melting and solidification, respectively. It can maintain the PV temperature during the full stage of heat transfer [19]. Several numerical and experimental studies have added microencapsulated phase-change material (MPCM) to water in a PVT model to reduce the panel temperature [20].

One of the keys that have been considered as a problem is the low thermal conductivity of PCMs, which provides greater cooling and maintains better thermal control of the board [21]. Nanomaterials were added to PCMs to obtain improved thermal properties compared to the original materials. The type and concentration of nanoparticles in nano-PCMs indicated their thermal characteristics. Furthermore, with the use of selective paint over the solar cell, it was observed that heat loss was greatly reduced, and air was passed as a coolant in a designed stream with ease and greater safety. PVTs provide low thermal conductivity and specific heat, resulting in low efficiency. This is why most studies have used nanoscale fluids and water as a medium for heat transfer [22]. The addition of nanoparticles to the liquid increases the thermal efficiency of the PVT system [23]. A number of research projects were applied to test the effectiveness of nanoparticle concentration and appropriate size in the main liquid [24].

The current paper's main objective was to study the research activities of previous studies in the literature for the cooling techniques used in thermal PV systems based on PCMs. This research is organized as follows: Section 2 introduces the PVT/PCM hybrid system. Section 3 presents the PVT system with nanofluids. Section 4 presents PV modules integrated with PCMs. Section 5 highlights PV modules integrated with a PCM and fins. Section 6 introduces the PCM/concentrated PV system. Section 7 reports the building-integrated PV using PCMs (BIPV). Section 8 presents the conclusion and future scope.

## 2. PVT/PCM Hybrid Systems

Hybrid systems (PVT) can produce electricity and heat in one compact system. The overall efficiency of these systems is higher than those of PV and thermal collectors separately and is the total thermal and electrical performance, and it can be improved through integration with PCMs. The squandered heat of the PV panel is first taken by PCMs in the form of latent heat, resulting in a decreased surface temperature of the PV, which leads to improved electrical efficiency. Figure 3 represents a schematic of a PVT/PCM system.

Depending on the type of coolant used to cool the system, hybrid PVT/PCM systems can be classified into two types: air-based hybrid PVT/PCM systems, shown in Figure 4, and water-based hybrid PVT/PCM systems, shown in Figure 5.

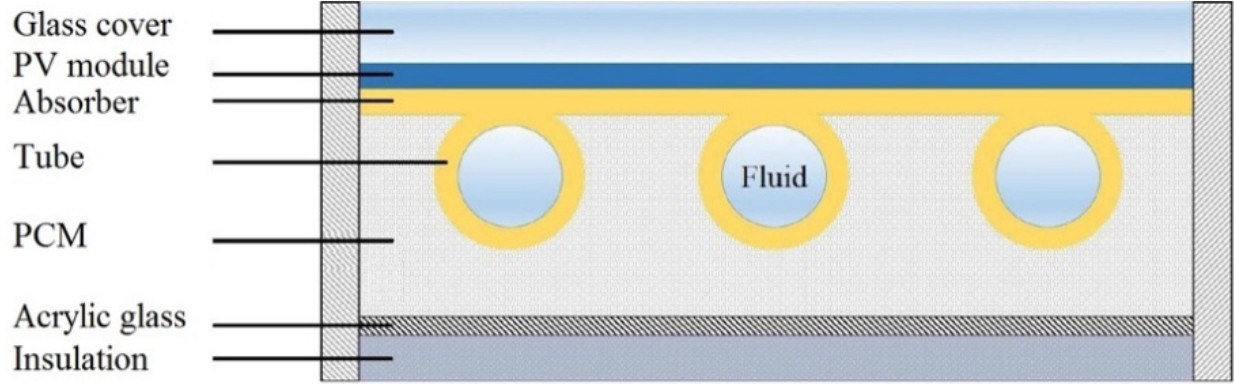

**Figure 3.** Diagram of an integrated PVT/PCM system [25].

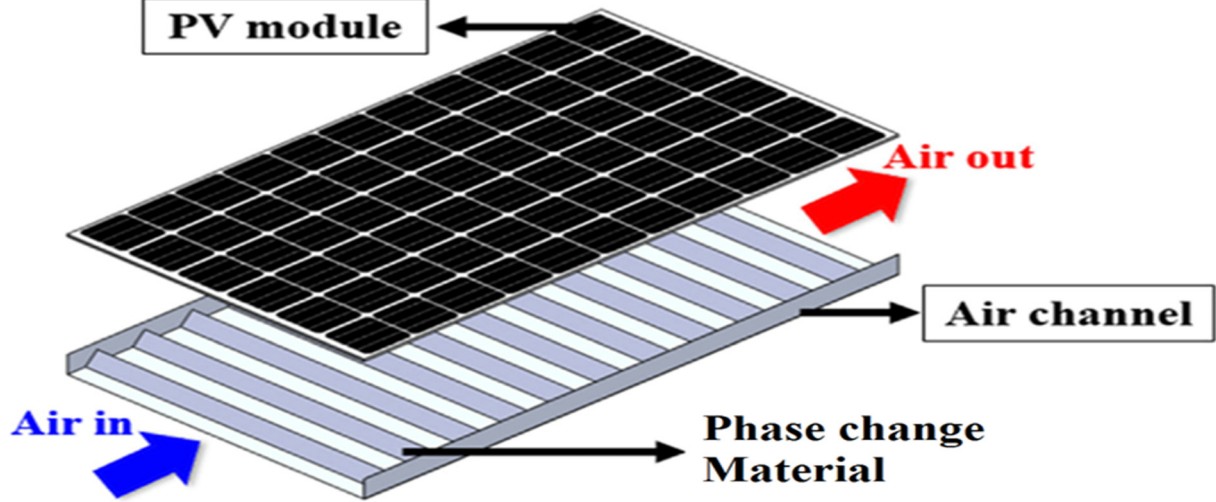

**Figure 4.** Diagram of an air-based hybrid PVT/PCM system.

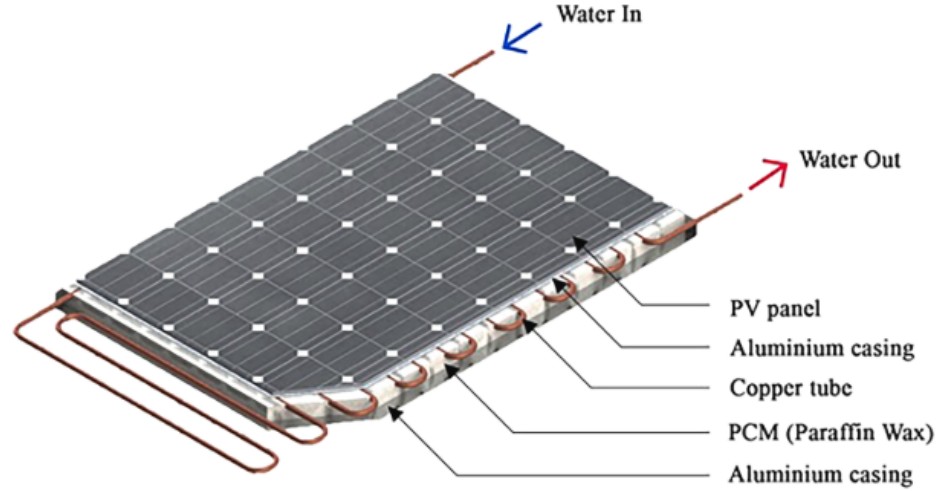

**Figure 5.** Diagram of a water-based hybrid PVT/PCM system.

### 2.1. Air-Based Hybrid PVT/PCM System

Ho et al. [26] used a microcapsule containing a PCM connected to the solar cell surface from back to form a microencapsulated PCM (MEPCM)/PV floating module (Figure 6). The capabilities of such a system to monitor the temperature and power production performance of PV modules were investigated using numerical simulation. During summer, different melting points (30 °C and 28 °C) and thicknesses (5 cm and 3 cm) of PVs were simulated under various environmental conditions. A layer with a thickness of 5 cm and a melting temperature of 30 °C had the best PCM result. The handled PV had a mean generation efficiency of 19.61% and a regular unit area generation capacity of 4743.29 kJ/m$^2$. Compared to that of untreated PV power generation, the capacity of treated PV power generation improved by 2.1%.

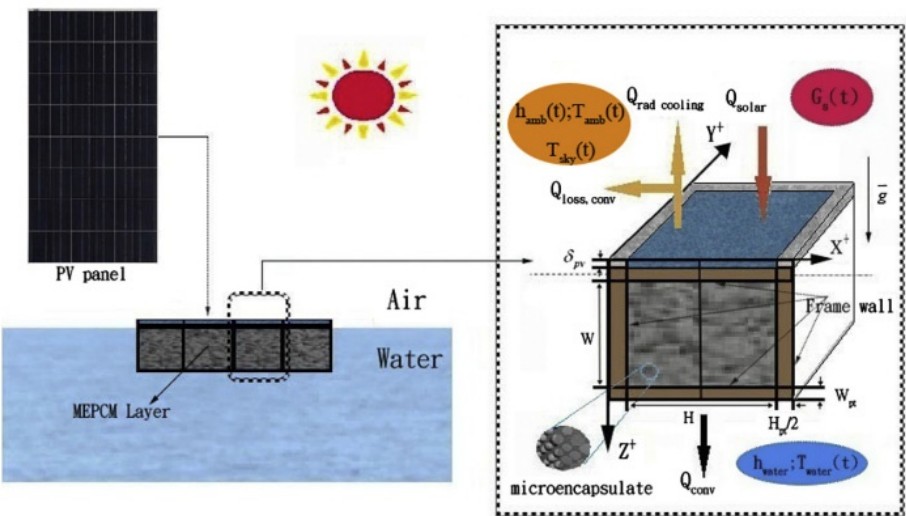

**Figure 6.** Diagram of the hybrid PVT/PCM system presented by Ho et al. [26]. Adapted with permission from Elsevier.

Ren et al. [27] investigated PCM performance for a PVT collector's thermal energy storage (TES) system using the Taguchi method (Figure 7). PVT systems can produce electricity and heat power throughout the day; the thermal power produced can be saved in the PCM TES and used for heating through the night. The Taguchi method was used in the simulation design in addition to the variation analysis for data analysis. The suggested system thermal performance was estimated in terms of advantageous energy saved in the TES. The two studies demonstrated that the phase-change material maintained a difference between the entry and exit temperatures of air of 2 °C.

An investigation into the incorporation of the phenomenon of natural convection with phase-change material below the surface of a PV panel for extracting the panel's heat was conducted by Akshayveer et al. [28]. The PCM organized the photoelectric temperature, and its efficiency increased. Experimental data were used to validate the digital model that was developed for the PVT/PCM system. In order to analyze the thermal and electrical properties of the PV/PCM systems and the air-integrated PVT/PCM system, with an incident solar irradiance of 800 W/m$^2$, a comparative numerical study was conducted. Reductions of 25% and 35% in the system photovoltaic temperatures of the PV/PCM and air PVT/PCM systems, respectively, increased the electrical efficiency of the photovoltaic cell in the PV/PCM and air PVT/PCM systems by 14.12% and 19.75%, respectively.

Furthermore, Xu et al. [29] analyzed the power conversion performance of an experimental system of PVT combined with PCM under the true external climate of Shanghai. Fatty acid PCM was used for the PV panel cooling. Five cases representing different thermoregulation of the PVT/PCM system were analyzed to improve the performance of the system. Noticeable temperature stratification was observed in this study for PCM in the

solar collector, even when using metal fins. The variation of PV temperature could be well reduced by PCM; the overall energy efficiency of the PVT system with complete thermoregulation was found to be higher than those of the two systems without thermoregulation by 5.4% and 22.2%, respectively.

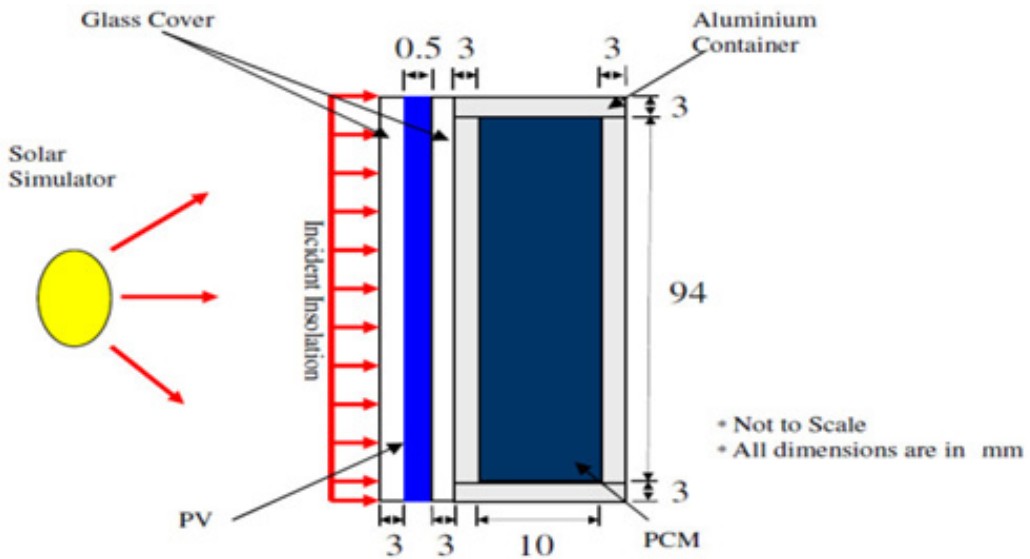

**Figure 7.** Diagram of the experimental setup presented by Ren et al. [27].

Fu et al. [30] presented the performance of a multilayer PVT system with PCMs experimentally and numerically (Figure 8). A detailed mathematical model of the heat transfer process and PVT system operational efficiency was updated. The temperature variation of the PVT system was simulated and verified by means of measurements based on an external experimental system. According to measurements and calculations performed using MATLAB software, the characteristic temperature and efficiency of the PVT system were predicted. The study found that the average efficiency for electrical of a PVT system could increase by about 1% by using a heat exchanger with PCM devices. Meanwhile, the thermal efficiency would be increased due to the long-running time. The PCM combined with 15% graphite improved the overall efficiency by 25.2%.

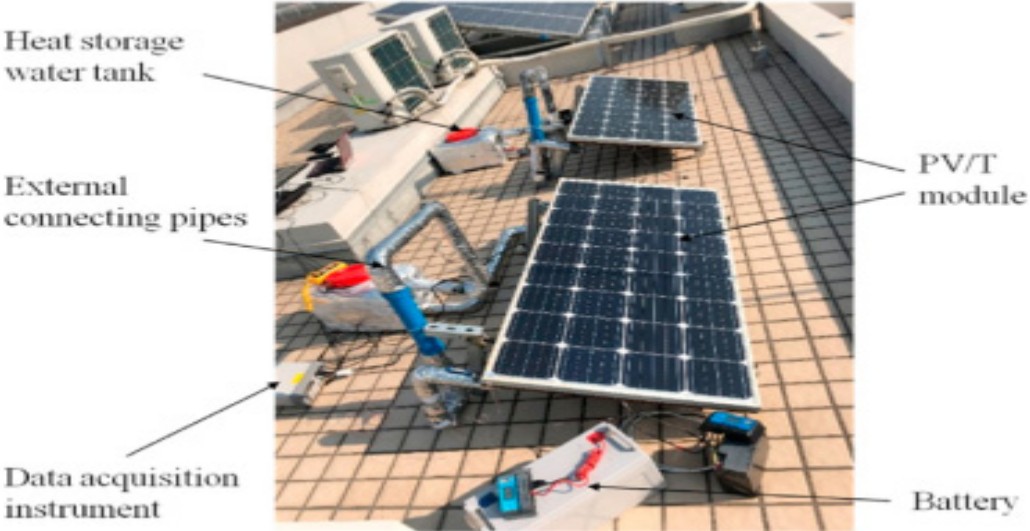

**Figure 8.** Diagram of the experimental setup presented by Fu et al. [30]. Adapted with permission from Elsevier.

Su, et al. [31] studied the effectiveness of PCMs with an air-based PVT solar compiler via numerical emulation. The results showed an overall efficiency enhancement by approximately 63% as compared to the decreased PCM mode. This resulted in a maximum PV temperature decrease of 7.4 °C for the PVT/PCM system when the thickness of the PCM was 3 cm. The system produced 10.7% higher overall efficiency without the phase-change material.

Maiti et al. [32] developed a model that stored cold energy in PCMs through the beginning of the morning when the ambient temperature was low. The air was cooled down after it passed out of the PCM container, as it flowed at a slight velocity to absorb the excess heat wasted by the photovoltaic panels, resulting in a decreased temperature during the day. This raised the electrical efficiency of PV panels. A new heat sink composed of PCM and mineral foam (porosity) that was used to cool down photovoltaic concentrators (CPV) with a solar concentration (CR) ratio of 20 was investigated to provide an optimal cooling system.

The effects of porous PCM systems were studied by J. Duan [33] for three groups of porosity ($\varepsilon$ = 80%, 90%, 100%) and four heights (H = 0.5x, 1.0x, 2.0x, 3.0x) to improve the electrical efficiency of CPV units. The results showed that metal foaming with high thermal conductivity incorporated into a PCM with high latent heat could significantly increase the cooling effect on a CPV unit as compared to pure PCM as a heat dispersant. A porous phase-variable material cooled the CPV, and the electrical efficiency of the solar cell increased as the porosity decreased. Temperature variation of the PVT system was simulated and verified by means of measurements based on an external experimental system. Moreover, the length of time that a CPV could be kept at a constant temperature decreased as the porosity decreased. The high porosity of a PCM was influenced by the cooling effect and electrical efficiency of the solar cell. When the porosity was the same, increasing the height (H) of the PCM porosity cavity from 0.5x to 1.0x raised the electrical efficiency and power productivity by about 50%, but a further height increase from 1.0x to 2.0x had little effect on improving electrical efficiency. When the porosity was less than 100%, increasing the height from 2.0x to 3.0x slightly reduced the electrical efficiency.

To optimize the solar PVT collector, an experimental investigation for an air-based complex with a TES and PCMs was performed by Lin et al. [34] using a PVT/PCM system, as shown in Figure 9. The experimental results showed an ideal airflow rate for the PVT/PCMs under a given solar irradiance to achieve the maximum overall system efficiency. At this airflow rate, the overall system performance improved from 37.6 to 40.2%, and the total daily utilization rate of latent TES power rose from 13.3 to 79.5%.

*2.2. Water-Based Hybrid PVT/PCM System*

To boost the efficiency of PV panels, a pilot analysis of different PV systems under various environmental conditions was performed. Thermoelectric plates, a water-based PVT system, and a PVT system with phase variable content (RT-30 paraffin wax) were used by Preet et al. [35]. The effects of three mass flow rates of 0.013 km/s, 0.023 kg/s, and 0.031 kg/s on electrical and thermal efficiency were studied in this experiment. Compared to the thermoelectric plate obtained at a mass flow rate of 0.031 kg/s of water during the day, the highest increase in electrical efficiency was 10.66 with the water-based PVT and 12.6 with the water-dependent PVT/PCM. The average increase in electrical efficiency with water-based PVT was around 23%, and with the PVT/PCM, 30%.

Results numerically simulated by using various water mass flow rates (0.25 kg/s, 0.5 kg/s, 1 kg/s, 5 kg/s, 10 kg/s) were used by Zhou et al. [36] to assess the effects of water mass flow and entry temperature on PVT/PCM output and examine the inlet temperature values for the flow of water. The cell temperature decreased, and photoelectric efficiency was improved. In addition, the influence of the inlet water temperature was greater than than that of flux on the performance of the PVT (Figure 10).

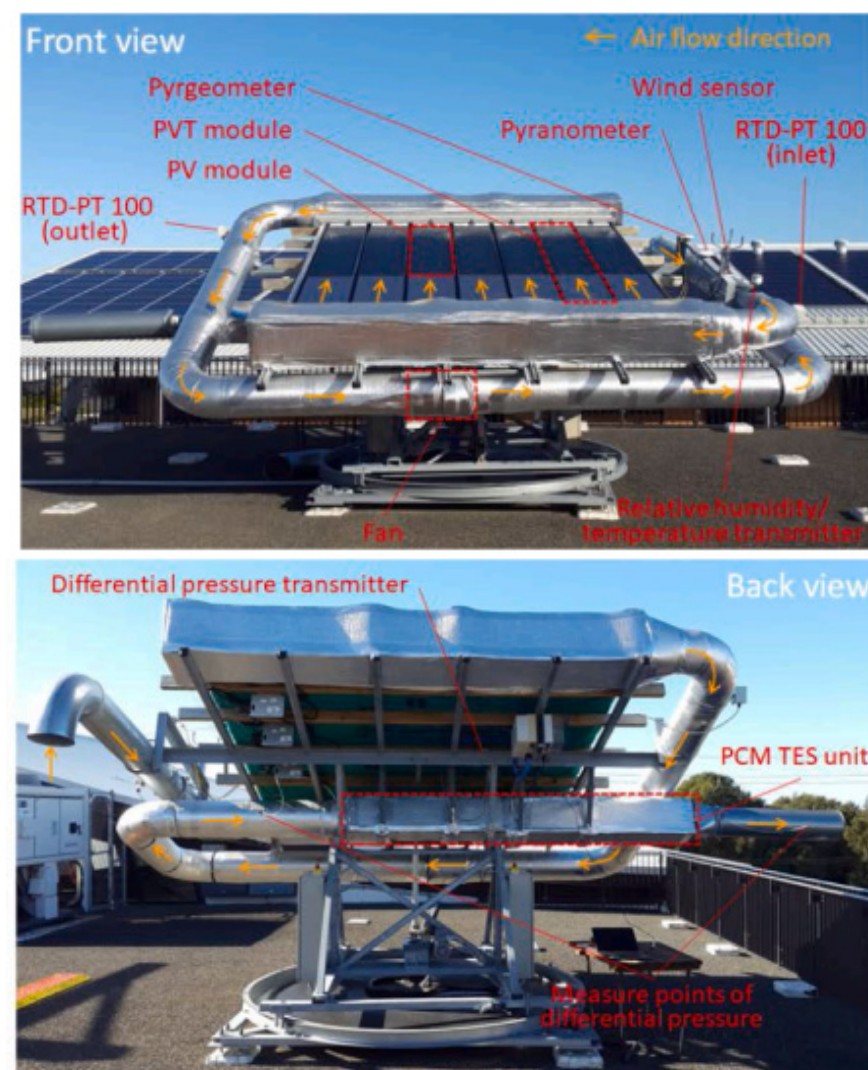

**Figure 9.** Diagram of the experimental setup presented by Lin et al. [34]. Adapted with permission from Elsevier.

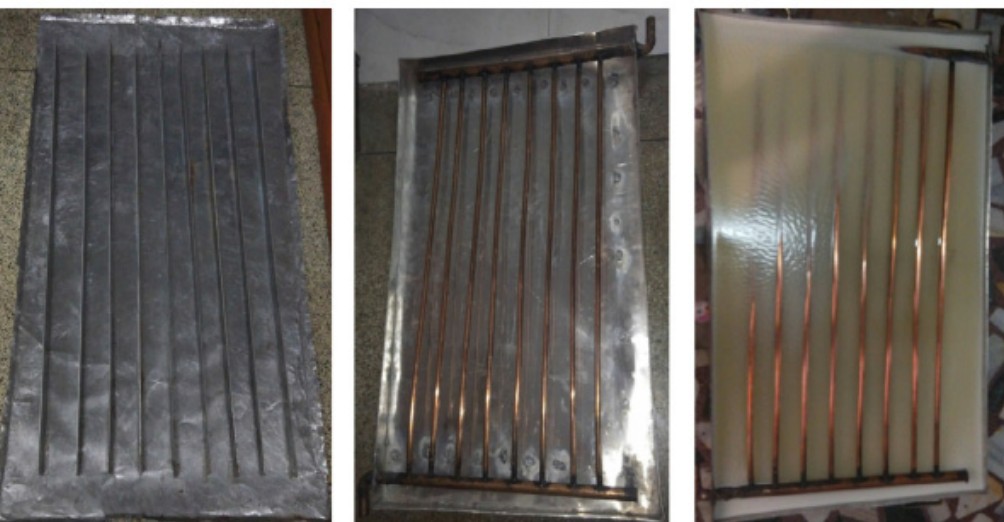

**Figure 10.** Diagram of three types of absorber plate used by Zhou et al. [36].

Lari and Sahin [37] designed a nanoscale cooled PVT system with a heat tank based on PCMs to meet part of the electrical and thermal requirements of a residential building in the desert climate of Saudi Arabia. A comparison was made with the performance of a nanoscale liquid-cooled PVT system without thermal storage and a noncooled PVT system, and an economic evaluation of the proposed system was conducted. Comparison of systems with and without thermal storage showed that the introduction of a thermal PCM battery led to an 11.7% improvement in electrical performance as compared to that of the noncooled PV system. The addition of thermal storage allowed coverage of up to 27.3% of the residential heat load along with 77% of the residential electrical load. The initial energy-saving performance of the PVT/PCM system improved by 14% in an analysis performed in an interior environment managed with 800 W/m$^2$ irradiances and a water flow rate of 0.15 m$^3$/hour. By calculating the power output and temperature of PV plate, the average energy efficiencies of PVT/PCM and PVT photovoltaic systems could often be compared.

The integral of the phase-change material with PVT was observed to effectively minimize heat loss into the environment. When solar radiation was less strong or unavailable, the heat contained in the PCM could be discharged into the operating fluid, extending its service life to the targeted house. These results indicated that incorporating PCM into a PVT system improved system efficiency. In the Kottayam District of India, Maatallah et al. [38] investigated a PVT/PCM/water device under various weather conditions. The overall efficiency of PV and water-based PVT/PCM panels was compared in experiments. The system was tested within various outside ambient conditions, and it was discovered that the PCM integration enhanced thermal performance and overall performance by 26.87% and 40.59%, respectively, with a 17.33% boost in electrical efficiency as compared to that of the conventional PV panels shown in the Figure 11.

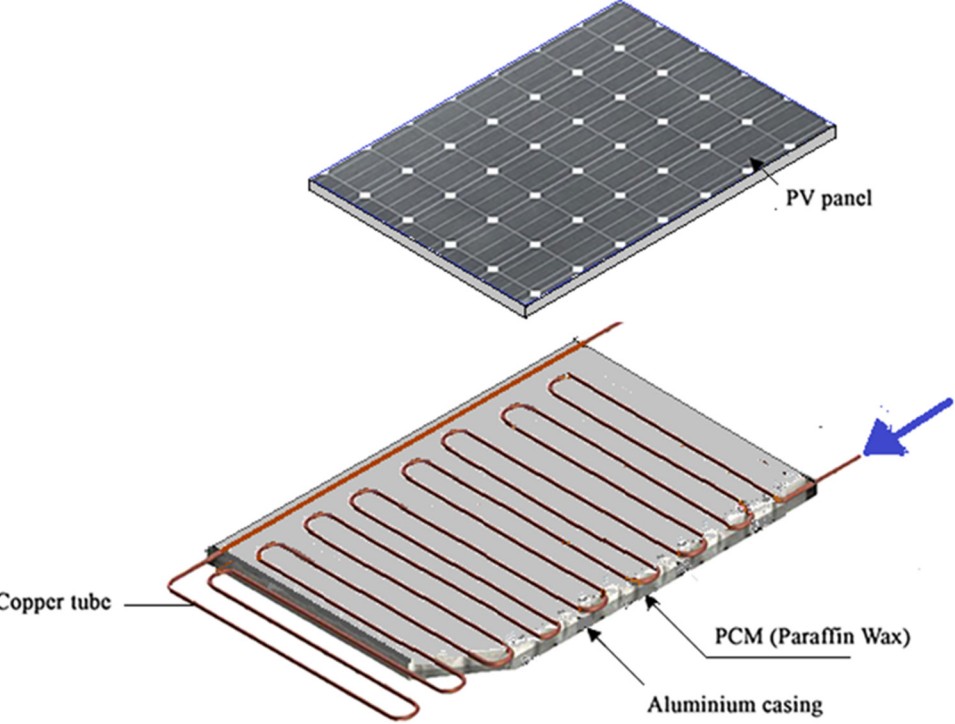

**Figure 11.** Diagram of the water-based PVT system studied by Maatallah et al. [38].

Hedayati-Mehdiabadi et al. [39] designed a double-slope solar pool with a PCM and built a PVT collector for fresh water in the evening (Figure 12). The system was checked for fresh production and energy efficiency of winter and summer samples in Zahedan, Iran. Water productivity increased by 10.6%, and energy efficiency increased by 27% and 2% on 6 July on 23 December, respectively, which increased the mass of brine from 20 kg to

30 kg. Freshwater production decreased by about 4.8% during the day but increased by 7.43% in the evening. The highest power generation values were 120 watts and 50 watts, respectively, on 6 July and 23 December.

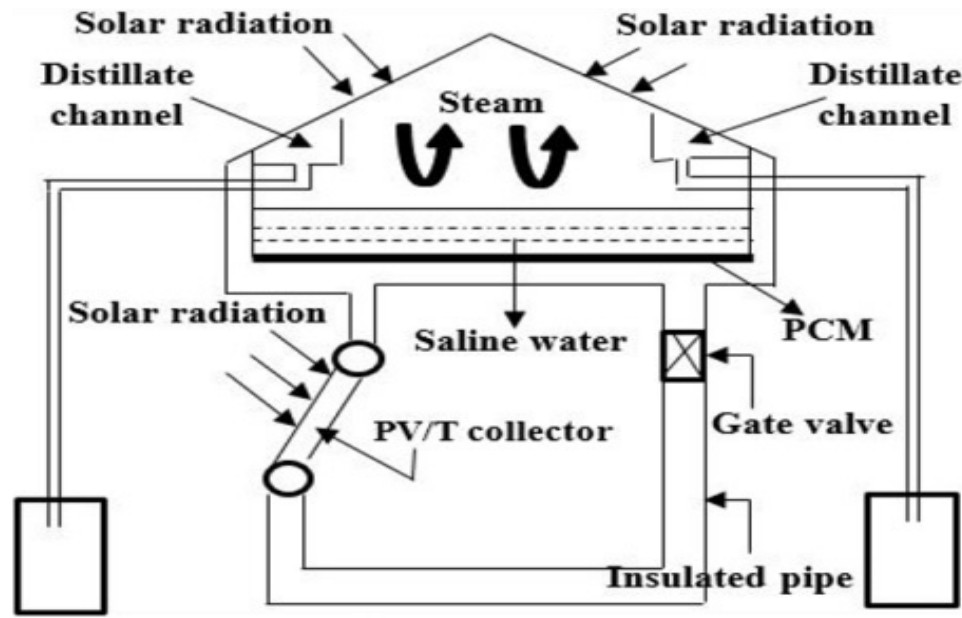

**Figure 12.** Diagram of the system studied by Hedayati-Mehdiabadi et al. [39]. Adapted with permission from Elsevier.

PVT and PVT/PCM systems are examples of modern thermal complexes that were engineered to improve heat transfer and efficiency. Numerical research in three dimensions was carried out by Fayaz et al. [40] and checked using an experimental study into the conditions of maintaining the entry water temperature and ambient at 27 °C and solar radiation at 1000 watts/m$^2$, which resulted in volume flow rates ranging from 0.5 to 3 L/min. The experiment was carried out in a laboratory setting with regulated operating parameters and passive cooling. The PVT had a gross electrical efficiency of 12.4% and 12.28% as calculated numerically and experimentally, respectively. The PVT/PCM also achieved 12.75% and 12.59% electrical reliability (numerically and experimentally, respectively). There were two types of solutions, numerical and experimental, under which the electrical efficiency of the PVT system was shown to have increased by 9.2% and 10.13%, respectively. Electrical efficiency improvements of 12.91% and 12.75% were achieved with the PVT/PCM in numerical and experimental terms, respectively.

Al Imam et al. [41] investigated PVT/PCM performance with a compound equivalent capacitor (CPC) for comparison on clear and partially cloudy days, as shown in Figure 13. The overall efficiency of the system increased about 10% more for clear days than for partially cloudy days. The temperature rise with PCM was slower. A very high mass flow rate of water under constant temperature was used with the PCM, which led to an increase in the heat discharge, and thus, the electrical efficiency increased. When a low temperature of water with a constant mass flow was used, the electrical efficiency also increased.

Experimental research was conducted by Xu et al. [42] on the performance of a PV solar heater for a combined system with a PVT/PCM. A solar collector filled with PCM was used, as shown in Figure 14, which contained rectangular metal fins to enhance heat transfer and to cool the PV. A water container was also used to improve the overall efficiency, and detailed comparisons were made. The results indicated that using PCM in the solar collector could greatly reduce the temperature fluctuations of the PV panel and improve PV efficiency. The overall efficiency of the suggested system was approximately 85%.

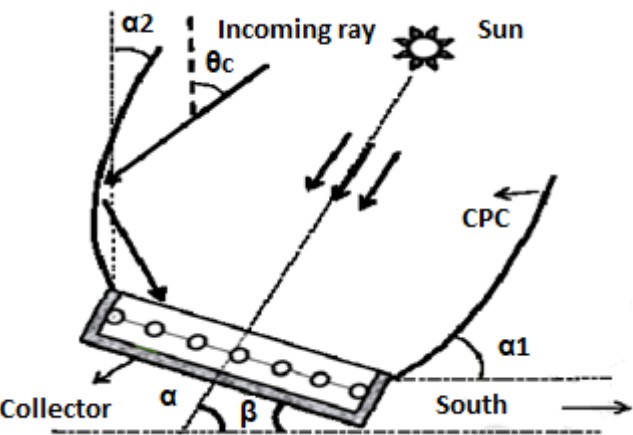

**Figure 13.** Diagram of the system studied by Al Imam et al. [41]. Adapted with permission from Elsevier.

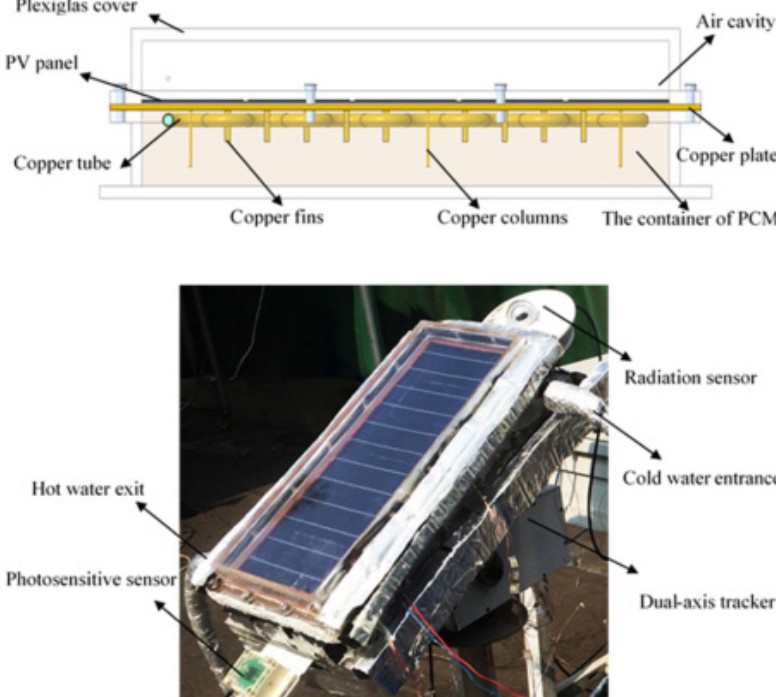

**Figure 14.** Diagram of the experimental setup studied by Xu et al. [42]. Adapted with permission from Elsevier.

Experiments using a cold eutectic phase-change material (C-PCM) to reduce the temperature of a PV unit were conducted by Velmurugan et al. [43]. Ethyl alcohol and lauryl alcohol were combined in 75:25 ratio on the rear surface of the PV module to achieve a melting temperature of 21.75 °C and a latent fusion temperature of 199 J/g. C-PCM boxes with thicknesses of 3 cm and 5 cm were attached, with the 5 cm box exhibiting greater heat transfer, lowering the TPV by a maximum of 10.3 °C. The average voltage was increased by 2.85% because of the drop in TPV, and the average output power was also improved by 2.8%. The PCM boosted average output ratios by 72.63% and electrical efficiency by 10.09%. According to the findings, C-PCMs might be a viable alternative to H-PCMs for cooling the PV units of commercial PCM machines that are costlier than eutectic PCMs. Qasim et al. [44] studied the effect of the fins on the performance of a PVT solar collector integrated with a PCM. The PCM alone without fins reduced the temperature of PV cells by a mean of 22.9 °C. The addition of fins resulted in maximum PV module temperature reductions

of 23.7 °C, 24.1 °C, 24.9 °C, and 26.5 °C with 2 fins, 5 fins, 8 fins, and 11 fins, respectively, around 12:30. Compared to naturally cooled PV panels, the power production improved on average by 12%, 15%, 18%, and 21%, respectively. In comparison, the efficiency of electrical conversion increased from 10.2% to 10.9%, 11.3%, 11.6%, 11.8% and 12.2%, respectively. The arrangement of 11 fins for the PV/PCM was the most efficient.

Touati et al. [45] performed a numerical study by using ANSYSflunet15 to charge and discharge stored solar thermal energy (in a TES system) in a PCM by using latent heat from a storage unit containing the PCM. The improvement in heat transfers between PCMs and the fluid that represented water induced by adding fins to the volume was studied in various ways. The fin geometry reduced the discharge time, as the overlapping fins provided better performance than the embedded fins. It also played between charging and discharging from the thermal point of view because of the difference in density (natural convection). Yuan et al. [46] conducted experiments to compare the performance of a PVT/PCM system and a PVT with an ordinary water-based straight pipe. The results showed that the electrical efficiency of the PVT without PCM was lower than that of the PVT/PCM. The thermal efficiencies of the two systems were 42.3% and 44.5%, respectively. Moreover, the temperature of the PVT with the PCM was lower because of the heat storage ability of the PCM, which allowed for PVT operation at a lower temperature.

A PV module, PVT collector, and PVT collector integrated with PCM thermal collector were combined to create a system that produced energy, retained heat, and heated water in outdoor conditions by Browne et al. [47], as shown in Figure 15. The efficiency of different configurations was compared (with those of the same system without the PCM or heat exchanger and the PV module alone). It was discovered that the temperature reached by the water was around 5.5 °C higher in the PVT/PCM system than in the PVT system without PCM. The PCM was an effective way of storing and removing heat for later use in a PVT system for other applications, such as water heating. Compared to that of a device without a PCM, PCM was shown to increase the thermal energy production from a photovoltaic module by up to seven times. It must be confirmed that the phase shift temperature happened at night for the PCM to solidify and store the most heat during the day. Table 1 shows a summary of the studies reviewed for PVT cooling by using PCM.

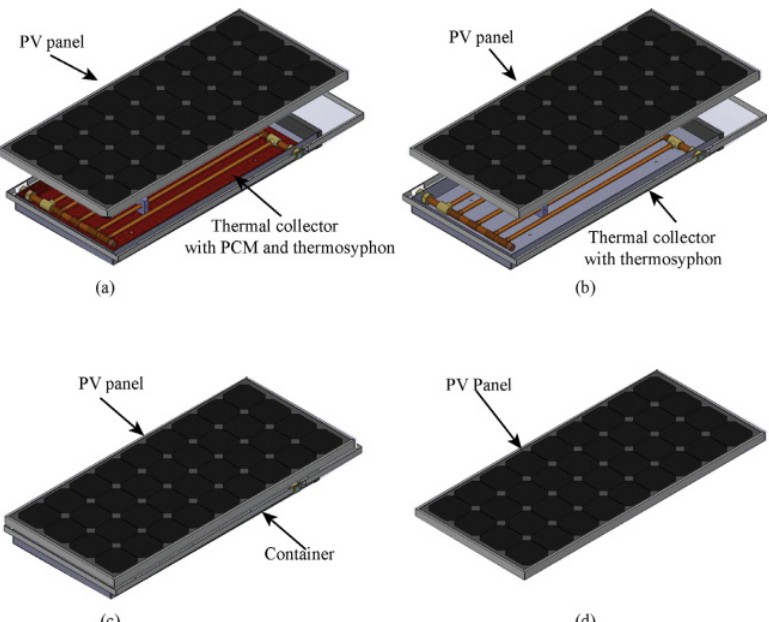

**Figure 15.** Diagram of the different designs: (**a**) PV/T/PCM, (**b**) PV/T, (**c**) PV with container, and (**d**) PV, studied by Browne et al. [47]. Adapted with permission from Elsevier.

**Table 1.** Findings of the studies reviewed for PVT cooling using PCM.

| Reference | Location | Parameter | Type of Study | Major Findings |
|---|---|---|---|---|
| G. Ménézo and G. Julien, 2017 [48] | Lyon, France | Fully wetted absorber PVT collector's thermal and electrical performance | Numerical study | PCM integration in a water PV absorber improved the execution of the system in terms of electrical and thermal parameters |
| Hachem et al., 2017 [49] | Lebanon | Photovoltaic cells using pure and combined PCMs | Experiments and transient energy balance | The electrical efficiency of the PV module was improved by an average of 3% when a pure PCM was used and by an average of 5.8% when a mixed PCM was used. |
| A. Hasan et al., 2017 [50] | United Arab Emirates | PV/PCM | Yearly energy performance | In hot weather conditions, the PV/PCM system raised the annual PV electrical energy yield by 5.9% |
| A. Hasan et al., 2014 [16] | China | PV/PCM | Energy and cost saving | At high temperatures and in high-solar radiation environments, PV/PCM systems were financially feasible. |
| Indartono et al., 2016 [51] | Indonesia | PV performance by using yellow petroleum jelly as PCM | Experimental study | Efficiency and power were increased. |
| Kazemian et al., 2019 [52] | China | PVT/PCM | Numerical investigation | The electrical and thermal energy efficiency of the PVT/PCM system increased as the thermal conductivity of the PCM increased. |
| Khanna et al., 2017 [53] | India | PV/PCM | Numerical study | As tilt angle increased from 0 to 90°, the PV efficiency increased from 18.1 to 19% by using the PCM, and efficiency was improved from 17.1 to 19% |
| Maatallah et al., 2019 [38] | Kottayam, India | PVT/PCM/water | Experimental study | The thermal efficiency, electrical efficiency, and overall efficiency improved by 26.87%, 17.33%, and 40.59%, respectively, as compared to those of the traditional PV panel. |
| Nouira and Sammouda, 2018 [15] | Tunisia | PV/PCM | Numerical study | Dust deposition density reduced electrical power. A rise in wind speed led to an increased heat loss. A wind azimuth angle increase caused an increase in the operating temperature of the PV module. |

**Table 1.** *Cont.*

| Reference | Location | Parameter | Type of Study | Major Findings |
|---|---|---|---|---|
| Preet et al., 2017 [35] | India | PVT system with and without PCM | Experimental investigation | Decreasing the PV panel's temperature increased its output power, i.e., electrical yield, and improved the production of electricity. |
| Simón-Allué et al., 2019 [54] | Spain | PCM influence on different models of PVT collectors | Experimental study | Distribution of heat output improved, generating up to 30% of the full thermal power after sun exposure was removed. |
| Smith et al., 2014 [55] | United Kingdom | PV energy output enhanced by PCM cooling | Global analysis | Better results were seen, whereby an optimal PCM melting temperature was chosen for the place in question, and the PCM melted completely during the day. |
| Waqas et al., 2017 [56] | China | Thermal behavior of a PV panel integrated with PCM-filled metallic tubes | Experimental study | An efficiency increase of up to 3% was observed. The fin effect was observed to cool the PV panel, as the PV panel was kept at a lower temperature. |
| Yang et al., 2017 [13] | China | Comparison of PVT/PCM and PVT systems | Experimental investigation | Thermal efficiencies of the PVT and PVT/PCM systems were 58.35% 69.84%, respectively. Solar electrical efficiencies of the PVT and PVT/PCM systems were 6.98% and 8.16%, respectively. |
| Zhao et al., 2019 [17] | China | PVT/PCM | Year-round performance analysis | As compared to the reference PV system, 2.46% produced the largest year-round increase in electricity. The economic analysis revealed that the PV/PCM system might not be viable for actual use at this point without a substantial increase in PCM efficiency or the utilization of electricity-and-heat generation. |
| Browne et al., 2016 [47] | Ireland | PVT/PCM | Heat retention | PCM was seen to be an efficient way of retaining heat for later removal of heat. |

**Table 1.** *Cont.*

| Reference | Location | Parameter | Type of Study | Major Findings |
|---|---|---|---|---|
| Kazemian et al., 2020 [57] | Mashhad, Iran | Glazed and unglazed PVT system integrated with PCM | Experimental approach | The dual-use of glass cover and PCM in PVTs contributed to increased efficiencies. |
| Lin et al., 2021 [34] | Australia | Optimization of a solar PVT collector coupled with PCM thermal energy storage | Experimental investigation | The total system average efficiency rose from 37.6 to 40.2%, and the latent TES capacity average daily use ratio improved from 13.3 to 79.5%. |
| Malvi et al., 2011 [58] | United Kingdom | Combined photovoltaic solar–thermal system incorporating PCM | Energy balance | The PV output increased typically by 9%, with an average water temperature increase of 20 °C. |
| Qasim et al., 2020 [44] | Pakistan | Hybrid PCMs on thermal management of PV panels | Experimental study | 1-PCM configuration demonstrated improved performance only when 1-PCM configuration had lower-melting point PCM than those in 2-PCM configurations |
| Touati et al., 2017 [45] | France | Discharging from a multiple-PCM storage tank | Numerical study | The geometry of the fins reduced discharge time, as the staggered fins provided better performance than the in-line fins. |
| Xu et al., 2020 [42] | China | PVT/PCM | Experimental study | The findings showed that the use of the PCM in the solar collector greatly reduced PV panel temperature variations and increased the performance of the PV system. |
| Yuan et al., 2018 [46] | China | PVT/PCM | Numerical simulation and experimental study | The PVT with PCM and water-pipe-based PVT results for daily electrical efficiency were 12.1% and 11.9%, respectively; the thermal efficiencies of the two systems were 42.3% and 44.5%, respectively. |
| Bigaila and Athienitis, 2017 [59] | Canada | PVT air collector assisting a façade-integrated, small-scale heat pump with radiant PCM panel | Numerical study | As compared to the entire air system, drops in energy consumption of 14.5% and in heating power of 11.3% were achieved. |
| Lin and Ma, 2016 [60] | Australia | Taguchi–Fibonacci search method | Experimental study | The coefficient of thermal efficiency enhancement of the house increased from 45.54 to 72.22% relative to the results without optimization. |

### 3. PVT Systems with Nanofluids

Al-Waeli et al. [61] conducted a study to enhance the effectiveness and efficiency of photovoltaic panels. The PVT/PCM was created, and the nanoparticles, in this case SiC nanoparticles, were added to the working fluid and a PCM to improve the thermal conductivity that lowered the cell temperature. The tests were carried out on the compound shown in Figure 16 using a mass flow rate of 0.17 kg/s. The panels' temperature decreased at the peak time to 30 °C, and the open-circuit voltage increased. Therefore, the electrical efficiency rose to 13.7%, from 7.1%, and the thermal efficiency reached 72%.

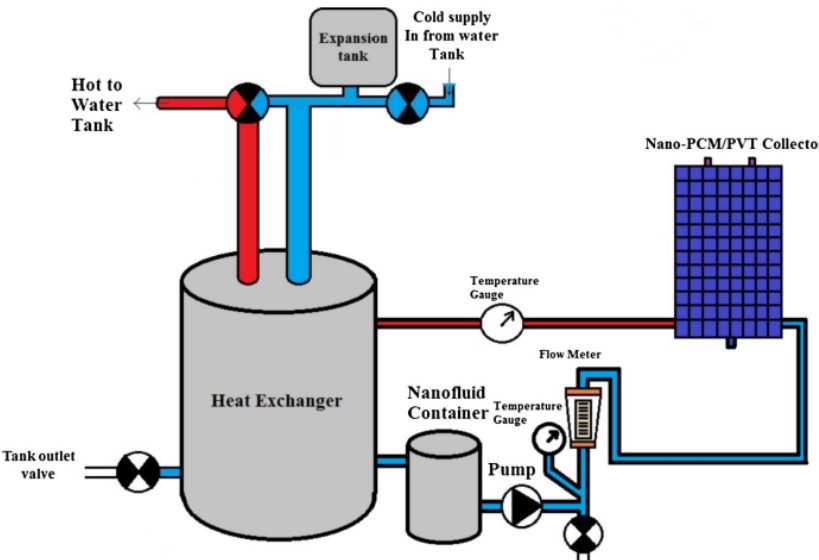

**Figure 16.** Schematic description of the experimental system studied by Al-Waeli et al. [61]. Adapted with permission from Elsevier.

Eisapour et al. [20] enhanced the efficiency of the thermal PV cell or module increased with the use of the corrugated tubes shown in Figure 17 as compared to the straight tubes. The results also demonstrated that among the various models of coolants, the nanocapsule-wrapped PCM coolant had the highest energy efficiency because of its high heat capacity and thermal conductivity. Through the use of the corrugated tube and the new cryogenic nanoscale, the electrical efficiency increased from 10.73 to 11.33%, the thermal efficiency increased from 58.56 to 63.74% for the typical PV thermal module, and the energy efficiency increased as compared to the typical PV thermal cell.

The multiwalled carbon nanotubes were considered nanoparticles because of their high thermal conductivity, which allowed a lower concentration of these nanoparticles, lower flow pressure reduction, and the depletion of pumping energy. A study on the efficacy of water/glycol-based nanoscale fluids as an effective cooling medium and PCM as a passive cooling medium was carried out by Naghdbishi et al. [62] as shown in Figure 18. The best efficacy of the PVT/PCM board was obtained to relatively enhance the aqueous nanoparticles' electrical efficiencies. The dispersion of the nanoparticles in the aqueous liquid led to an increase in the electrical and thermal energy efficiencies of up to 4.21% and 23.58%, respectively, as compared to pure water used as coolant.

Sardarabadi et al. [63] conducted a study in which nanofluid was used in deionized water to improve the average thermal output (of about 5%) for the PVT system. The electrical production rate was increased by more than 13% in the collector system based on PCM/nanofluid compared to that in the conventional PV module. An increase of about 9% in thermal efficiency was observed without any additional depletion of energy. Based on exergy analysis results, the simultaneous use of both nanofluids and PCMs for the cooling system increased the overall system energy efficiency by more than 23% compared to that of the conventional PV cell.

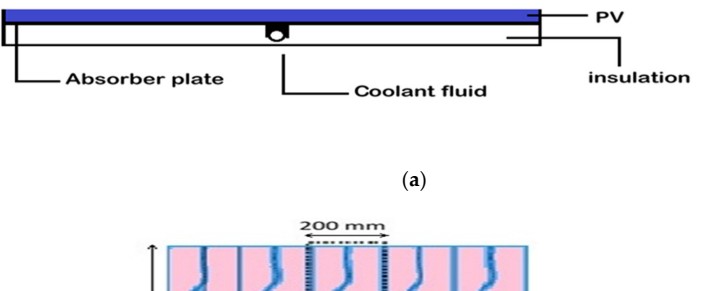

(**a**)

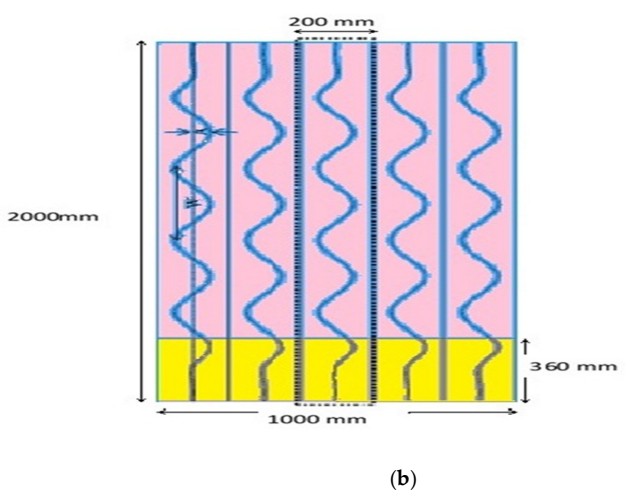

(**b**)

**Figure 17.** Diagram of the wavy tube with PVT system: (**a**) local cross-section and (**b**) front view.

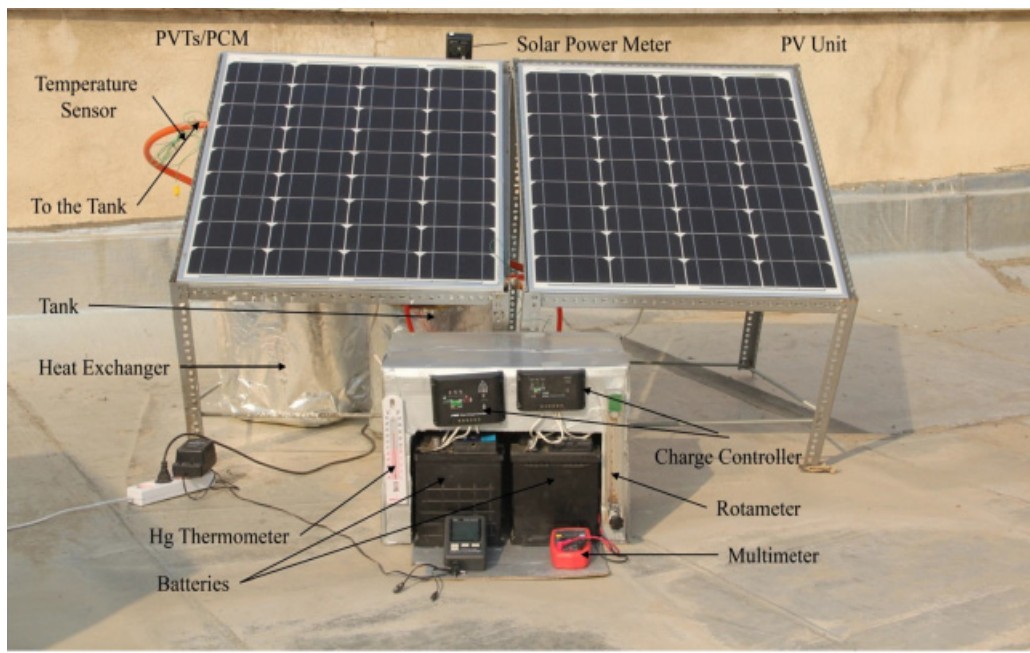

**Figure 18.** Photograph of the experimental facility studied by Naghdbishi et al. [62]. Adapted with permission from Elsevier.

A concentration of 2% nanoalumina particles was added to paraffin wax by Nada et al. [64]. The results showed that by using PCM in a pure state and PCM that contained alumina nanoparticles, the temperature decreased by 8.1 °C and 10.6 °C, respectively, compared to the PV panel without a cooling system. This decrease in temperature improved the efficiency by 5.7% and 13.2%, respectively. The heat decline was due to an increase in the PCM's thermal conductivity with the use of nanoparticles and thus an increase in heat transfer. Table 2 shows the summary of the studies reviewed herein on PVT cooling by using nanofluids and PCM.

**Table 2.** Summary of the studies reviewed herein on PVT cooling using nanofluids and PCM.

| Reference | Location | Parameter | Type of Study | Major Findings |
|---|---|---|---|---|
| Al-Waeli et al., 2017 [61] | Selangor, Malaysia. | Nanofluid- and nano-PCM-based PVT | experimental study | Increased the open circuit voltage from 11–13 to 20–21 V, the power rose from 61.1 to 120.7 W, the electrical efficiency rose from 7.1 to 13.7%, and thermal efficiency reached 72%. |
| Eisapour et al., 2020 [20] | | PVT systems using microencapsulated PCM nanoslurry coolant | Exergy and energy analysis | Higher performance energy and energy efficiencies due to higher thermic conductivity and heat capacity were found. |
| Ho et al., 2016 [6,26] | Taiwan | PV integrated with double water-saturated MEPCM layers | Numerical simulation | The thermal and electrical efficiency of the MEPCM/PV module greatly improved. |
| Naghdbishi et al., 2020 [62] | Iran | MWCNT/water-based PVT/PCM | Experimental investigation | The thermal and electrical efficiencies increased up to 23.58% and 4.21%, respectively, as compared to pure water as coolant fluid. |
| Qiu et al., 2016 [65] | China | MPCM slurry-based PVT system | Experimental investigation | (1) Increasing the amount of slurry Reynolds resulted in improved solar thermal and electrical efficiencies, an increased drop in pressure, and decreased module temperature, and (2) increasing the concentration of MPCM resulted in decreased module temperature and an increased drop in pressure. |
| Abdelrazik, Saidur, and Al-Sulaiman, 2020 [66] | Saudi Arabia | PVT/PCM system using different combinations of nanoenhanced PCM | Thermal regulation | Increasing the loading of nanoparticles in a PCM provided better cooling and improved overall performance. |
| Sardarabadi et al., 2017 [63] | Iran | ZnO/water nanofluid and PCM in PVT systems | Experimental study | The simultaneous use of both a nanofluid and a PCM for the cooling system, based on the results of an exergy analysis, improved the system average exergy performance by more than 23% relative to that of a traditional PV module. |
| Tanuwijava et al., 2013 [67] | Taiwan | Thermal management performance of MEPCM modules for PV applications | Numerical Investigation | The microencapsulated PCM layer aspect ratio had important effects on the characteristics of heat transfer and overall thermal efficiency. |

## 4. PV Modules Integrated with PCMs

PV modules can be cooled through passive cooling using PCMs that absorb heat from the unit in the form of latent heat for solidification and fusion so that the temperature remains at a certain limit for a long time. This can improve better by combining different types of PCM, adding fins to overcome PCMs' low thermal conduction problems, or by using two PCM units simultaneously. The electricity production of PV modules can be increased by using thermal storage materials such as PCMs. Heat can be drawn in latent heat, thus dissipating the heat produced by the work [44]. Comparison between conventional PV modules and PV modules that use PCM for cooling indicated that those that use PCM had decreased operating temperature by 7 °C and less fluctuation. Furthermore, in Greece, as a photovoltaic module was kept at 25 °C, PCMs showed effectiveness in cooling photovoltaic panels when combined with PV [45,49].

The use of thermal energy storage materials, such as PCMs, can increase the electricity production from PV by dissipating the heat resulting from the high operating temperature during work [68]. Japs et al. [69] studied the effect of using variable-phase materials to cool PV panels compared with conventional PV panels. The panels' temperature was decreased by 7 °C by using PCM, and fewer fluctuations in temperature were observed, which gave more uniformity. In Greece, when the backs of PV panels were combined with PCMs, the PCMs effectively cooled the PV panels. The cell temperature was kept about 25 °C and was ideal for PV [70]. A review of a PV/PCM system using paraffin wax (RT28HC) in a cloudy atmosphere, at a maximum temperature of 35.6 °C and an average temperature of 14.4 °C, saw a decrease in the PV temperature. Because of the high latent heat of PCM (245 kJ/kg), the PV temperature remained nearly stable throughout the phase change process from 12:30 to 14:00. The simulation results also indicated a 7.3% rise in electricity production [71].

Brano et al. [72] presented a simple numerical sample for the PV/PCM system by exploiting the finite difference method and comparing it with the test system by using paraffin wax RT27 to verify the sample validity with different PV temperatures. The study measured and calculated an error of only 7% paraffin wax RT35 was used to maintain the PV module temperature. The cell temperature was reduced by 35 °C in the simulations and by 10 °C in the test conducted during a clear sunny day with a PCM layer of 2 cm thickness. The temperature was kept at a stable temperature (35 °C) for more than 5 h as mentioned by Mahamudul et al. [73]. To estimate the year-round energy-saving efficiency of a photoelectric phase change (PV-PCM) system of an extremely hot nature, a paraffin-based PCM was incorporated into the back of a PV panel. It was found that the PCM showed volatile efficacy in varying months during summer due to the high ambient temperature, as the PCM was not able to completely solidify during the night, which was reflected in the performance. In winter, the cooling performance degraded because the PCM could not receive enough heat energy to melt completely. In the moderate months, paraffin melted fully during the day because of the ambient temperature and sufficient radiation and solidified at night because of an adequate drop in temperature. The PCM finally achieved a 10.5 °C decline in PV panel temperature at medium peak time, which increased the PV panel production by 5.9% year-on-year [74]. Sarwar et al. [75] considered several angles ($-45°$, $0°$, $45°$, and $90°$) as well as different concentration ratios (CR = (5, 20)) when using a PCM ($CaCl_2$ $6H_2O$) with layers of different thicknesses (50 mm, 200 mm) on the efficiency of a PV. At a 45° slope, the PV's efficiency was at the maximum level (17%) because of the low average panel temperature. The hottest temperature was recorded at 90°, while the coldest was at $-45°$ because of the increased CR and the decreased thickness of the PCM layer. The CPV/PCM with a slope angle of 45° reached the lowest mean temperature with sensible compatibility against the solar cell temperature. These conditions ensured the highest electrical efficiency of the solar cells and helped to prohibit hot spots in the solar panel. M. Emam [76] conducted research on the effectiveness of varying solar radiation (1000 W/m$^2$, 750 W/m$^2$, and 500 W/m$^2$) on the temperature of photovoltaic unit that contained a paraffin wax (RT20) in an aluminum container by using a solar energy simulation system in the laboratory. On the back surface, the PCM had the highest latent heat absorption at the

highest radiation; the maximum temperature drop was obtained at 1000 W/m$^2$, followed by 750 W/m$^2$ and 500 W/m$^2$. Ceylan et al. [77] reported during mathematical molding that paraffin wax was very good at decreasing pane temperature. At the concentration ratio of CR = 3, the temperature was above 100 °C for a plate without a PCM, while it was between 80 and 100 °C for a plate with PCM, and the ambient temperature was 47 °C. Huang et al. [78] achieved passive cooling of PV cells by using phase-variable RT25HC and assessed its performance through a theoretical study. The inclination angle of the PV panel varied from 0 to 90°. The performance of the system was better with a tilt angle of 90°, as this angle increased the heat load inside PCM. The largest decrease in PV module temperature (19 °C) was achieved with the PCM, and the resulting energy and efficiency improvements were 11.1% and 11.8%, respectively.

The influence of PCM melting points (28 °C and 30 °C) and the influence of PCM layer thicknesses were tested during numerical simulation by Savvakis and Tsoutsos [79], and the results showed that the thickness of the layer (3 cm and 5 cm) was able to regulate the PV temperature at 30 °C and that the highest PV temperature was 30 °C. The power output was 2.1% at a maximum thickness of 5 cm; the 3 cm thick PCM resulted in a lower PV temperature decrease.

Senthil Kumar et al. [80] conducted research using copper, silicon carbide (SiC), and paraffin petroleum wax by integrating the PCM installed on the back of the PV module into an aluminum container in the second model. The phase-variable material's volume was 2870 cm$^3$, and its mass was 2.87 kg. Of this mass, 2.0 kg was pure paraffin wax, 0.574 kg was copper, and 0.287 kg was silicon carbide. The PV temperature increased from 38 to 52.9 °C, with an average of 43.4 °C, when the PV panel was without a PCM, and from 35 to 49.6 °C, with an average of 40 °C, when the PV panel was cooled with the PCM in its back surface. These figures indicated that common PCM use can reduce PV panel temperature by up to 4.5 °C and an average of 5.4 °C. This combined PCM reduced the PV panel temperature more than no PCM. The electrical output of the PV panels when using the combined PCM increased by a rate of 4.3%.

Savvakis et al. [81] evaluated energy performance under Mediterranean conditions. Integrated-PCM and conventional PV modules were developed, and two PMC types, paraffin wax RT27 and RT31, were used. The maximum temperature was reduced by 6.4 °C and 7.5 °C, respectively. Masses of of 260 g of RT27 and RT31 increased power generation by 4.19% and 4.24%, respectively. The increase in the photoelectric conversion efficiency induced via PCM integration ranged from 2.86 to 4.19%.

In Alexandria, Egypt, a new, simplified one-dimensional mathematical model was developed by Elsheniti et al. [82] to predict the temperature of a PCM in contact with PV cells, with near-computational fluid dynamics (CFD) accuracy while greatly reducing calculations and time, based on estimation of the thermal conductivity equation enhanced by thermal currents within the PCM during the melting and solidification processes. A CFD model was also developed to simulate PV/PCM performance. The validity of the two models was confirmed by demonstrated published test data and good compatibility. Comparisons were made between both models with different inclination angles (from 0° to 90°), which were illustrated during the high-melting density period and for all inclination angles of 0.74% and 1.78% for the lowest and highest ratios, respectively.

A new technique to improve the efficiency of solar PV panels by using PCMs and aluminum panels such as TCE was proposed by Rajvikram et al. [83]. They used two 5 W photoelectric panels; one was combined with an aluminum plate at the back of the panel. The panel was compared to one combined with a naturally ventilated plate without PCM and aluminum. This treatment was tested with external exposure to direct sunlight for three years. It was experimentally verified that the aluminum plate at the back of the panel improved the conversion efficiency of the panel by an average of 24.4%. With a decrease in average temperature by 10.35 °C, the electrical efficiency of the plate increased by 2%. The maximum decrease in temperature was 13 °C for the first day and 7.7 °C for the second day.

## 5. PV Modules Integrated with a Phase-Change Material and Fins

When a photovoltaic system is turned on, its temperature increases while its electrical output reduces. The heat generated during the process can be extracted by applying a PCM to the rear of the PV plate, which can pull heat in latent heat of solidification and fusion. This leads to a large drop in PV temperature and enhances PV performance. Fins should be used within a PCM container to maximize heat transfer. Studies have shown that after the PCM is fully melted, its heat removal rate decreases. This leads to a rise in PV temperature. It is known that phase-variable materials suffer from reduced thermal conductivity; hence, fins used to improve thermal performance reinforce them. A two-dimensional digital model was developed to mimic the use of a PCM with fins inserted in the PV panel to lower the temperature of photovoltaic cells at various angles from 0 to 90°. A paraffin wax PCM (RT25) was put in close contact with a PV plate and covered by aluminum plates from both faces and with fins included. Expected temperatures were compared with numerical and experimental data from previous studies, and better agreement was reached. The temperature of the PV plate rose with an increasing tendency, and a slight inclination (less than 45 °C) was made for better cooling of the plate [84].

Khanna et al. [85] calculated the optimal depths of PCM reservoirs for various degrees of solar irradiance as well as the effects of distance between the successive fins affected. The effect of fin height and optimal fin measurements were also studied. The analysis revealed that the best depths for PCM containers were 2.8 cm for $\sum IT = 3$ kWh/m$^2$/day and 4.6 cm for $\sum IT = 5$ kWh/m$^2$/day. To sustain a lower PV temperature (25 cm), the optimal fin thickness was 2 mm, and the optimal fin length was when fins came into contact with each other. The reservoir foundation, PV, PV/PCM, and finned PV/PCM models were also compared for a PV/PCM device (no fins). The most suitable thickness for the PCM reservoir was 2.3 cm for $\sum IT = 3$ kWh/m$^2$/day and 3.9 cm for $\sum IT = 5$ kWh/m$^2$/day. In another study, a temperature-finned PV plate was filled with a PCM and fixed with different tilt angles. The studied system was designed as a two-dimensional rectangular container filled with a paraffin wax PCM (RT25) and placed between two aluminum plates. The front side was exposed to a constant heat flow of 1000 W/m$^2$ for 2 h. Four shapes were compared: a nonfinned PCM enclosure, a PCM enclosure with a full-width central fin, a half-width fin attached to an aluminum front plate, and a half-width fin attached to the back plate. The most efficient configuration for obtaining an acceptable temperature was the full-width fin connected simultaneously to the front and back panels. With this PV panel design, melting occurred to the PCM because of the natural convection being transferred from both sides early, with additional heat loss from the rear panel to the outdoor environment. Accordingly, low values of front and back panel temperatures could be maintained during a stability period of 80 min and with varying angle of inclination from 0 to 75° from vertical. The resulting effective temperature control was not related to the geometry of the fin [86]. In [87], the natural thermal behavior of a fuse for a PCM in a rectangular reservoir, which could be proper for cooling applications for photovoltaic panels, was investigated. Types and lengths of fin inserted inside the PCM reservoir and the heat transfer performance were estimated after the melting process, while emulation was performed by looking at three various aspect ratios (AR = 1, 2, and 4), three different Rayleigh type figures (Ra = 104, 105, and 106), two fin types (rectangular and treelike branched fin), and three length-to-height ratio of rectangular fin (0.3, 0.4 and 0.5). Though fins are useful for quick phase change, they reduced natural heat transfer by convection modification in the melted PCM once the melting process was completed [87].

## 6. Phase-Change Material/Concentrated PV System

The results of [41], which used a PCM compound parabolic center for a PVT solar collector, reflected that the collector's thermal efficiency ranged between 40 and 50% for a clear day and was about 40% for a semiclear day. The overall efficiency of the PVT ranged between 55 and 63% on a clear day and about 46 and 55% on a semicloudy day, while the value of the greatest loss was around 3 W/m$^2$ for a clear day and about 2.5 W/m$^2$

on a semicloudy day. However, if the same conditions were applied without a PCM, the temperature increased sharply, and then it was nearly constant [41]. Based on temperature and utilization of recovered heat to cool the low-concentration PV cells, the heat stored in the melt PCM was exchanged with the local supply water. The outdoor CPV/PCM/T system was evaluated in extremely hot weather conditions in the United Arab Emirates for three consecutive days. The power performance of the CPV/PCM/T system was compared to that of a common flat plate collector (FPC) side by side in the same location. Although the CPV/PCM/T produced less net power than the FPC, the production cost was 28% less than the cost of producing electric power [19]. Cui et al. [88] used a Fresnel lens to enhance the performance of a PVT collector integrated with PCM as shown in Figure 19.

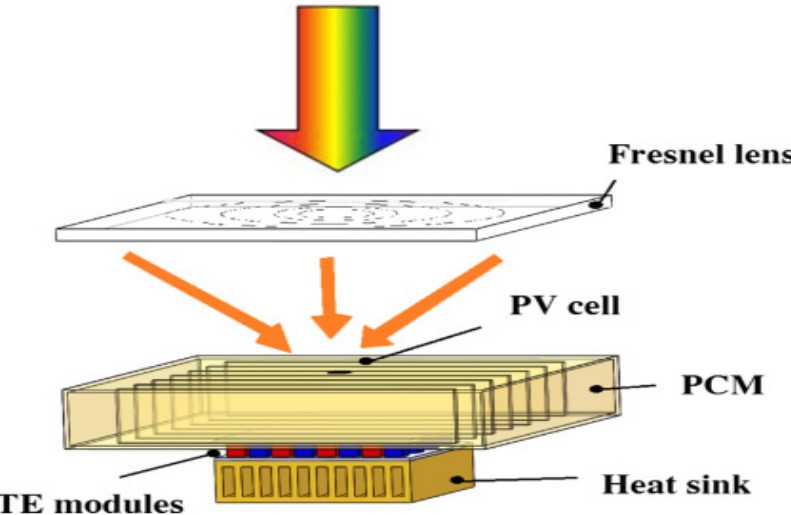

**Figure 19.** Details of the experimental setup studied by Cui et al. [88]. Adapted with permission from Elsevier.

The results of a detailed performance study on a PV/PCM/TE hybrid system showed that proper design of the system could effectively enhance the solar energy conversion efficiency. The angle of inclination of the CPV/PCM system had a large consequence on the desired time to fully melt the ephemeral shift in the average cell temperature and the local temperature coherence of the PV cell. These results indicated a significant decrease in the average solar cell temperature with increasing inclination; with an inclination angle of $-45°$, the maximum temperature range was obtained along with a sudden and unwanted change of cell spot temperature. Moreover, at a 45° inclination angle, the highest electrical efficiency of the cell was attained from the beginning to the end of the period until a complete melt occurred [27]. Table 3 shows a summary of the proposed models for concentrating photovoltaic thermal system cooling by using PCMs.

**Table 3.** Summary of the study-suggested models for concentrating photovoltaic thermoelectric system cooling using PCMs.

| Reference | Location | Parameter | Type of Study | Major Findings |
|---|---|---|---|---|
| Al Imam et al., 2016 [41] | Bangladesh | Compound parabolic concentrator and PCM in a PVT solar collector | Experimental study | The overall efficiency of PVT was between 55% and 63% for a clear day, the thermal efficiency ranged from 40% to 50% for a clear day. |
| Cui et al., 2016 [88] | China | Concentrating photovoltaic–thermoelectric system with PCM | Theoretical work | The findings revealed that the PV/PCM/TE system's efficiency was superior to those of single-PV panel and PV/TE systems. |

**Table 3.** *Cont.*

| Reference | Location | Parameter | Type of Study | Major Findings |
|---|---|---|---|---|
| Cui et al., 2017 [89] | China | A concentrated photovoltaic-thermoelectric system with PCM | Experimental investigation | Such a hybrid system had promising potential for the full-spectrum use of solar power. |
| Emam et al., 2017 [76] | Egypt | Inclined CPV/PCM system | Study and analysis | The angle of inclination of CPV/PCM system had a notable impact on the time taken to reach the full melting state, the transient difference of the mean cell temperature, and the uniformity of the PV local temperature. |
| Tabet Aoul et al., 2018 [19] | United Arab Emirate | CPV/PCM/T system with FPC | Experimental study | While the CPV/PCM/T generated less net energy (1527 kWh/m$^2$.day) than the FPC (1803 kWh/m$^2$-day), the production cost was 28% lower than that of the FPCC. |

## 7. Building-Integrated PVs (BIPVs) Using PCMs

A study was carried out to assess PCMs for use in integrated PV buildings to improve thermal regulation. The study discovered that using PCMs to regulate the temperature of BIPVs reduced the efficiency loss of temperature-dependent PVs. The thermal conductivity of both the PCMs and all PV/PCM systems and the thermal mass of the PCMs were found to be important factors in PCM thermoregulation. At a solar radiation intensity of 1000 W/m$^2$, a maximum temperature reduction of 18 °C was achieved in 30 min, with a temperature reduction of 10 °C being maintained for 5 h [90]. The effect of a PCM system for integrated photovoltaics on the energy efficiency of building in hot climates was studied by using the PV/PCM system as a component to improve energy efficiency with photoelectric cooling and reduce heat transfer inside. The PV/PCM system was developed by adding a PCM layer behind the PV. In the case of using the PCM, a lower temperature of the photoelectric plate and reduced heat loss was observed in the surrounding environment, as 47.7% of the incident radiation was stored as heat energy compared to 0.36% in the case of PV alone. Therefore, a rise in PV production of 7.2% was observed at the peak and 5% on average, with an improved cooling effect indoors of 9.5% at the peak and 7% on average during the day [91]. Through experiments and numerical simulations, a study was conducted on the use of a PCM to modify the temperature rise of a building's PV panel. With the PCM at 20 mm, the surface temperature was preserved at 36.4 °C for 80 min. For a PCM with a fusion temperature of 26.6 °C, the depth of PCM was 40 mm. With insolation of 750 W/m$^2$ and an ambient temperature of 20 °C, the temperature at the face surface of a two-fin system was maintained at less 33 °C for 150 min. A PCM depth of 30 mm inside the PV/PCM system without art could maintain the temperature on the surface of the PV system at below 35 °C [92].

The effect of PCMs on reducing the temperature increase in integrated PV cells in buildings was investigated. Two particular PCMs were used to mitigate the overheating of PV cells. Thermal performance was offered to improve the thermal conductivity of the PCM for different inner fin arrangements by using paraffin wax RT25 with inner fins. The temperature rise of the PV/PCM system could be decreased by more than 30 °C as compared to a single flat aluminum sheet rest through a phase change, showing that PCM materials constitute an active method to lessen the temperature rise in photovoltaics. Granular PCM GR40 could be used with the PV panel to reduce overheating, but the

thermal control was not as efficient as when solid and liquid paraffin wax PCMs (RT25) were used [93]. The thermal and photovoltaic solar collectors obtained the best total energy production by combining a 3.4 cm-thick PCM layer with a fusion point of 40 °C. In contrast to the electrical power of the fusion-point condition of 30 °C and no phase change in the case of multilayers, the largest variation was 16.12 watts at 12:00. This indicated that the electrical energy increased by approximately 13.6% by combining a PCM layer with a fusion point of 30 °C [94]. Tanuwijava et al. [67] investigated the thermal performance of MEPCM capsules for PV applications in conjunction with a building construction system under changes in solar irradiance. The results showed that aspect ratio (Am), which represented the width-to-height ratio of the PCM layer (capsules), importantly affected the thermal performance. The two states' general and heat-transfer properties examined with different fusion points (26 °C and 34 °C) were nearly the same. The use of an MEPCM stratum with Am = 0.277 increased the efficiency of the PV cells by approximately 0.1%, while the use of a MEPCM stratum with Am = 1 reduced the average stable efficiency by around 0.14%. A BIPVT air-based incorporated photovoltaic collector/thermal collector simulation in structure façades was performed with a small-scale air-to-water heat pump and a radiating plate with a PCM. For southwest-facing perimeter areas in a cold climate, the results showed that a 14.5% reduction in electrical energy demand could be obtained for these design days. In comparison, an 11.3% reduction in heating energy demand was obtained for the aforementioned designing days when comparing to all-air systems [59]. Table 4 summarizes the various studies reviewed herein for PVT cooling in buildings by using PCM.

**Table 4.** Summary of the studies reviewed herein for building photovoltaic system cooling by using PCM.

| Reference | Location | Parameter | Type of Study | Major Findings |
|---|---|---|---|---|
| Huang et al., 2004 [92] | Ireland | BIPV using PCM | Experimentally validated numerical model | The temperature moderation achieved will lead to considerable changes in the operating performance of photovoltaic facades. |
| A. Hasan et al., 2016 [91] | United Arab Emirates | In a hot environment, a PV/PCM device improving building energy efficiency | Experimental study | There was a 7.2% rise in PV power production at peak and 5% on average, along with an increase in indoor cooling effect of 9.5% at peak and 7% on daytime average. |
| A. Hasan et al., 2010 [90] | Ireland | PCM for improving the thermal control of a building-integrated PV | Experimental study | For 30 min, a mean temperature reduction of 18 °C was reached, while a temperature drop of 10 °C was sustained for 5 h at 1000 W/m$^2$ insolation. |
| Bigaila and Athienitis, 2017 [59] | Canada | PVT air collector assisting a façade-integrated, small-scale heat pump with radiant PCM panel | Numerical study | Compared to the entire air system, drops in energy consumption of 14.5% and in heating power of 11.3% were achieved. |
| Huang et al., 2006 [93] | United Kingdom | PCM in BIPV | Experimental evaluation | PCM was shown to be an effective means of minimizing temperature rise in PV systems. |

## 8. Conclusions and Future Scope

Photovoltaic panels can be cooled by using a passive approach through the use of PCMs. PCMs store the harmful heat generated by solar panels during the conversion process, which is not converted into electrical energy, in the form of latent heat. PCMs are integrated into the back surface of PVT panels so that the panel's surface can be preserved for a certain period with limited temperature. The use of a PCM regulates the temperature, improves the performance of the photovoltaics, and saves the energy and cost of the photovoltaic in the system. It also increases the life of photovoltaic cells via the lower operating temperature and the increase in the energy density of buildings integrated with the photovoltaic panels. The best application of these systems is in environments with radiation and high temperature, as their effectiveness is more evident in summer conditions than in winter conditions. More heat is absorbed in summer, which leads to improved efficiency more than in winter. The use of PCMs in cooling photovoltaic panels is considered an uneconomical solution because of the material's high cost, low thermal conductivity, and unreliable solidification and fusion behavior. Improvements can be made to PCMs to overcome the lower thermal conductivity through the combined use of PCMs, and they can also be improved and better controlled by combinations with different types of fin arrangements. Efficiency can be improved by using heat dispersants. In addition, adding nanoparticles to PCMs improves their thermal conductivity. This reduces the cell's temperature, but among the problems of this technique are its high cost and the problem of nanoparticle clumping. Therefore, more studies are needed in the future.

**Author Contributions:** Conceptualization and methodology, M.M.A.; investigation, O.K.A.; resources, O.M.A.; writing—original draft preparation, N.T.A.; writing—review and editing, S.J.Y.; visualization, A.N.; supervision, M.A.; project administration, and funding acquisition, A.F.A. All authors have read and agreed to the published version of the manuscript.

**Funding:** This project is funded by King Saud University, Riyadh, Saudi Arabia.

**Acknowledgments:** Research Supporting Project number (RSP-2021/323), King Saud University, Riyadh, Saudi Arabia.

**Conflicts of Interest:** The authors declare no conflict of interest.

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
