# Peer review of "Photovoltaic Thermal Collectors Integrated with Phase Change Materials: A Comprehensive Analysis"

_electronics, doi:10.3390/electronics11030337_

Round 1

Reviewer 1 Report

Authors attempt to review on the thermal collector by PV application with details integrated with PCM. PCM and its derivatives is discussed. Anyway, authors did not clearly review on mechanism, novelty, and how is the thermally improvement by PCM.Other than that, below are some suggestions for authors as reference.

  1. In abstract, authors should further explain what is PCM?
  2. Some of the words are more than one space. Example: mentally friendly. _Different types of the PV system…
  3. In introduction section, authors should include the basic structure of PCM in PV application. Authors must further explain how PCM helps and also problem statement of review topic.
  4. Page 2, line 49, the third one, what is meaning by the third one?
  5. Page 2, line 99, phase change materials term no need to mention again since it is repeatedly in previous text.
  6. Section 4: introduces the hybrid system for PVT/PCM, should bring to earlier section.
  7. Lack of diagram for illustration. Exp: hybrid system for PVT/PCM, air-based hybrid PVT/PCM system, water based-, and etc…
  8. Authors must include more the PCM diagram for every respective application in order to attract readers attention.
  9. Some of the references are out of date. Suggest that revise and update the references more than 5 years.
  10. What is the definition of RT24, RT25, RT27, RT-30, RT28HC, and so on. It is not clearly explain.
  11. Plenty of the uncommon term and didn’t explain well. Exp: what is the CFD?
  12. In conclusion, authors should include what is the thermal improvement after apply PCM. How is the reliability against life time improvement?

   Overall format: found that some uneven line spacing. Authors also should check carefully the entire manuscript for the repeated term and further clarify the common term to let readers more understand.

Author Response

I would like to thank you for your useful feedback to my manuscript and appreciate the constructive comments of the reviewers. I have carefully read through the reviewer comments, and provided accurate constructive responses

Reviewer 2 Report

In this paper, the authors present a review of photovoltaic thermal collectors integrated with phase change materials. Although the review is comprehensive, there are some writing issues that should be addressed before the paper can be considered for publication. Attention should also be paid to the paragraphs to improve coherency and flow of the paper. Additional figures should be included to strengthen the review.

  1. For a review study, the abstract should be comprehensive providing some detailed measures of performance and the key research gaps that have been identified.
  2. There are several English language problems that require significant revision and editing of the paper. I will not point out specific sentences, the entire paper would benefit from revisions by an English editor.
  3. In the introduction, first paragraph should comprehensively highlight the benefits of solar energy, state why solar PV is gaining increased attention. Then the sentence starting with ‘However, renewable harvesting by solar PV…..’ is vague.
  4. The paper itself is incoherent, there are several paragraphs and sentences that are not connected to each other. For example, the second paragraph comes before the challenges associated with solar PV are presented. This paragraph ideally presents a solution to a problem that should be clearly discussed. PCM is one of the solutions, there are other solutions that should be reviewed and highlighted. Then describe how the use of PCM is beneficial compared to the other solutions.
  5. The sentence ‘PV cells are semiconductors…. ‘ should probably come before we even talk about PCM. Generally, the entire paper should be significantly re-arranged. Then you have in this paragraph a sentence ‘Passive cooling by exploiting PCM….’ This is what you already discussed in paragraph 2.
  6. After this, the studies that have considered PCM in PVT systems should be reviewed and summarised here.
  7. Several ideas and sentences are repeated throughout the paper. First sentence under section 2 was already used, then this paragraph and the few sentences of next paragraph are almost saying the same thing.
  8. Several references in the last paragraph on page 3 are not linked to the citations. Moreover, this and other paragraphs are too long, they should be reduced with each paragraph focusing on a single idea and well supported by the following sentences.
  9. The citation style should also be streamlined and made consistent, it is preferable to use author names instead of ‘the author in [ ].
  10. There should be significant incorporation of figures to show some of the concepts. This is a main concern for such a review not to have figures. Figures would aid the audience not familiar with the concepts being described.
  11. A significant aspect that should be included is the economic competitiveness and technical maturity of the use of PVT-PCM systems. Are there any commercially implemented systems? To a home user, as the complexity of the system increases, the likelihood of adopting such a system will decrease. Provide a section before the conclusion to discuss this.
  12. Conclusion, the PCM does not actually store the incident rays, but the heat generated during the conversion process? Also, the conclusion should be improved significantly. What improvements in performance have been shown in the reviewed studies? Providing some numbers would be helpful. What about the cost considerations in using PCM in PVT systems?

Author Response

(The authors gave the same response as above.)

Round 2

Reviewer 1 Report

After one round of revised, authors have do the appropriate revision. Now the entire manuscript is looks better.

Author Response

I would like to thank you for your useful feedback to my manuscript and appreciate the constructive comments of the reviewers
